# Thinking Deeper With Recurrent Networks: Logical Extrapolation Without Overthinking

## Abstract

Classical machine learning systems perform best when they are trained and tested on the same distribution, and they lack a mechanism to increase model power after training is complete. In contrast, recent work has observed that recurrent networks can exhibit logical extrapolation; models trained only on small/simple problem instances can extend their abilities to solve large/complex instances at test time simply by performing more recurrent iterations. While preliminary results on these "thinking systems" are promising, existing recurrent systems, when iterated many times, often collapse rather than improve their performance. This "overthinking" phenomenon has prevented thinking systems from scaling to particularly large and complex problems. In this paper, we design a recall architecture that keeps an explicit copy of the problem instance in memory so that it cannot be forgotten. We also propose an incremental training routine that prevents the model from learning behaviors that are specific to iteration number and instead pushes it to learn behaviors that can be repeated indefinitely. Together, these design choices encourage models to converge to a steady state solution rather than deteriorate when many iterations are used. These innovations help to tackle the overthinking problem and boost deep thinking behavior on each of the benchmark tasks proposed by Schwarzschild et al. (2021a).

## 1 Introduction

Humans solve complex logical reasoning problems through the process of logical extrapolation – they assemble simple logical primitives into complex strategies. For example, a person taught to prove simple lemmas can in turn prove more complex theorems simply by expending more cognitive effort. Neural networks have achieved great success at pattern matching tasks, often exceeding human performance, but they lack the ability to solve complex reasoning tasks in a scalable way. Recently, deep thinking systems (Schwarzschild et al., 2021b) have been proposed as a way to represent and learn scalable reasoning processes using recurrent neural networks. The word 'thinking' in this context refers to sequential processing. These systems train recurrent models (networks that recycle parameters between layers) to solve reasoning problems. Unlike traditional feed-forward models, which are limited in the complexity of problems they can solve by their finite depth, the effective depth of recurrent models can be expanded after training simply by iterating the recurrent unit for longer. When trained properly, thinking systems learn scalable algorithmic processes for solving problems. After training to solve small/easy problem instances with few recurrent iterations, the

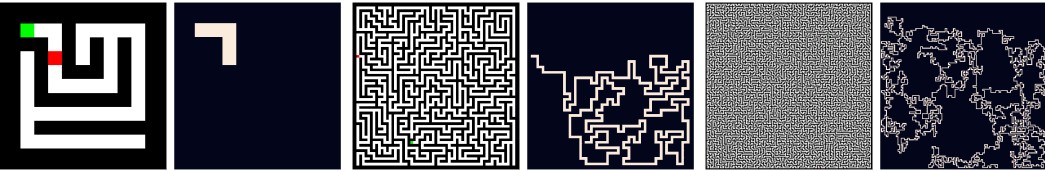

Figure 1: Our improved thinking network trained on $9 \times 9$ mazes and its solution (left two plots) autonomously synthesizes a scalable algorithm. By running this algorithm for longer, it solves $59 \times 59$ (center) and $201 \times 201$ (right) mazes without retraining. Standard architectures, and even existing primitive thinking models, fail to tolerate this domain shift.

system is then extended to run for more iterations at test time. In doing so, the system can achieve logical extrapolation, and solve problems of greater difficulty than those in the training set.

To date, the level of logical extrapolation observed in thinking systems has been quite modest. For example it has been demonstrated that a system trained on $9\times9$ mazes can extrapolate to solve a $13\times13$ maze. These systems fail to achieve greater extrapolation because of the *overthinking* problem; recurrent systems, when extended too far outside their training regime, often deteriorate and fail to produce interpretable outputs.

In this work, we design purpose-built neural architectures and specialized training loops to make it possible to train systems that do not suffer from overthinking and instead converge to a fixed point when iterated for thousands of iterations. By doing so, we are able to build thinking systems that exhibit extreme logical extrapolation behaviors, and leap from solving small/simple training problems with tens of iterations to solving large and complex problem instances at test time using hundreds or even thousands of iterations.

We experiment on the benchmark problems made available by Schwarzschild et al. (2021a) for measuring extrapolation behavior. These tasks include computing prefix sums, finding optimal solutions for two-dimensional mazes, and solving chess puzzles. We modify model architectures and propose a novel training routine for recurrent networks that significantly boosts performance on all tasks in the benchmark suite. Additionally, we demonstrate that existing models that have shown positive logical extrapolation on these datasets specifically are susceptible to overthinking, a problem that is largely overcome by our new architectures and training loops (Schwarzschild et al., 2021b).

Our contributions can be summarized as follows.

**I.** We provide a recurrent architecture for logical extrapolation in which the problem input is concatenated directly to the feature stack of certain layers in the recurrent thinking module. This prevents the problem instance from being forgotten if deep features become noisy, corrupted, or lossy.

**II.** We develop a new training routine that incentivizes recurrent networks to make incremental improvements towards a solution, improving the feature representation on each iteration. This training process removes information about how many times the recurrent module has been applied, which in turns prevents the network from learning iteration-specific behaviors (e.g., a special behavior for iteration five and another for iteration six) rather than scalable behaviors than can be iterated indefinitely to yield extrapolation.

**III.** We analyze the overthinking problem and show that our models overcome this phenomenon.

Our improvements in performance on the easy-to-hard benchmark datasets (Schwarzschild et al., 2021a) can be categorized in several ways. First, our models yield uniformly higher accuracy across the most difficult tasks used for testing in previous work. Second, we show that our models can extrapolate to much harder/larger examples than are considered in previous work, where the prior methods generalize poorly, if at all. Lastly, we show that our models do not forget solutions in settings where previous models overthink.

## 2   RELATED WORK

The work in this paper conceptually builds upon the systems recently studied by Schwarzschild et al. (2021b), which showed that simple recurrent architectures, when trained to solve various reasoning problems, can then perform logical extrapolation while their feed-forward counterparts cannot. In this section, we contextualize this approach amongst prior work on algorithm learning, adaptive neural models, and logical extrapolation.

**Algorithm learning** describes models that learn recurrent processes from data. Early works on this topic study the ability of recurrent neural networks (RNNs) to process input strings of arbitrary lengths (Gers & Schmidhuber, 2001; Schmidhuber et al., 2007). More recently, Graves et al. (2014) introduced neural Turing machines designed to mimic programmable computers, and Kaiser & Sutskever (2015) propose a parallel version inspired by massively parallel graphical processing units. These methods, and improvements to them, show promising results on bit string to bit string tasks, including copying inputs and adding integers, and even demonstrate the ability to generalize from shorter training strings to longer ones at testing (Graves et al., 2014; Kaiser & Sutskever,

2015; Freivalds & Liepins, 2017). These methods are based on classical RNNs so the amount of computation they perform is directly linked to the length of the input string, which prevents them from executing more or less computation independently of the input size (or the difficulty of the problem). Moreover, classical RNNs are often trained incrementally to produce one bit at a time, rather than synthesizing an algorithm for solving an entire problem end-to-end. This makes it difficult to apply them in situations where the solution to a problem cannot be decomposed into incremental parts (e.g., chess).

Constraint satisfiability problem (CSP) solving networks disentangle the amount of work from the input size (Selsam et al., 2018). Specifically, message passing neural networks can execute more passes to solve harder CSPs (Selsam et al., 2018). These systems are specific to the problem of constraint satisfaction for boolean expressions, but they are an early demonstration of scalable algorithmic behavior.

**Adaptive neural networks** are designed to expend varying amounts of computation on different inputs, thus overcoming the limitation of classical RNNs. Self delimiting neural networks use one neuron to determine when to stop updating the hidden state in RNNs, and in doing so, they perform more or less computation for each token in an input sequence (Schmidhuber, 2012). Adaptive compute time (ACT) is an algorithm that provides RNNs with a halting unit which estimates the probability that computation should continue. This algorithm penalizes 'ponder time' during training to encourage the network to solve problems quickly (Graves, 2016). Eyzaguirre & Soto (2020) exhibit strong performance on visual question answering by introducing a differentiable version of ACT. Adaptive transformer-based language models also exist, notably Universal Transformers and Depth-Adaptive Transformers, which utilize ACT to determine the work required for each input (Dehghani et al., 2018; Press et al., 2021). Similarly, iterative residual networks like NAIS-Nets repeat blocks in stages and perform well on image classification (Ciccone et al., 2018). All of these works test their methods *in-distribution*, i.e. where the training and testing data are sampled from the same distribution; logical extrapolation outside the training domain is not considered.

**Logical extrapolation** denotes the task of generalizing to test sets which comprise more computationally complex samples than the training data. Nuamah (2021) claims that "neural network models with end-to-end training pipelines ... cannot, on their own, successfully perform algorithmic reasoning," and instead proposes a hybrid hand-crafted and learned approach. Similarly, Palm et al. (2018) propose recurrent relational networks, which operate on graphs by iteratively passing messages. They also claim that classical architectures lack the inductive bias to reason about relationships between objects. Several recent works call these claims into question. Schwarzschild et al. (2021b) employ recurrent networks based on weight-sharing architectures, which can be made deeper at test time independent of the input size (Schwarzschild et al., 2021c). These systems were shown to exhibit logical extrapolation behavior in several domains. More details on these methods are discussed in Section 3 as our algorithms build on these directly. Banino et al. (2021) reformulate the halting unit in ACT leading to a probabilistic model with improved performance called PonderNet, and importantly their method outperforms ACT on logical extrapolation tasks for prefix sums.

The thinking systems proposed by Schwarzschild et al. (2021b) depart from classical recurrent networks for (e.g.) text, which learn from step-by-step supervision to produce output tokens one at a time. In contrast, thinking systems autonomously synthesize a scalable algorithm end-to-end with no supervision over what each algorithmic step should produce. For this reason, they can potentially be applied to solve complex problems that are difficult or impossible to decompose by hand.

## 3 METHODS

We begin with some terms and definitions. We study networks that share weights across blocks of layers during training. For example, instead of four distinct residual blocks, a single residual block is repeated four times (see Figure 2 for a graphical depiction). At test time, networks trained this way can be made "deeper" to extend their compute budget simply by repeating the block more times. We refer to the number of convolutional layers applied in a recurrent network as its "depth", and this quantity grows as the number of recurrent iterations increases. The number of feature maps produced by each convolution is referred to as its "width."

More formally, let $r$ be a function representing a recurrent block, e.g. a ResNet block (He et al., 2016), and let $r^n$ denote $n$ recurrences of that function, e.g. $r^2(x) = r(r(x))$. Let $\phi$ denote a feature map, or an output of $r$, and let $\phi_n = r^n(x)$. Finally, we consider an initial "embedding function" denoted by $p$, which projects an input instance into feature space, and also a final "output head" denoted by $h$, which maps features to outputs. A Deep Thinking (DT) network with $m$ iterations of the recurrent block can then be expressed as follows.

$$f_m(x) := h(r^m(p(x))) \tag{1}$$

In our systems $p$ comprises a convolutional layer followed by a ReLU, $r$ is a single four-layer residual block, and $h$ is a set of three convolutional layers with a ReLU after the first and second. We fix $m$ during training and compute gradients for optimization by backpropagating through the unrolled network. Then, $m$ can be increased for testing, allowing these networks to increase their processing power and solve larger and harder problems. Below, we consider two modifications to standard approaches for achieving improved extrapolation: modified architectures and modified optimization.

### 3.1 ADDING RECALL TO THE ARCHITECTURE

When humans think for a long time to solve a problem, we often stop to reread the question or review the task at hand. We modify thinking architectures to periodically recall the input exactly. We incorporate this capability into architectures by concatenating the input problem to the features output from each instance of the recurrent block.

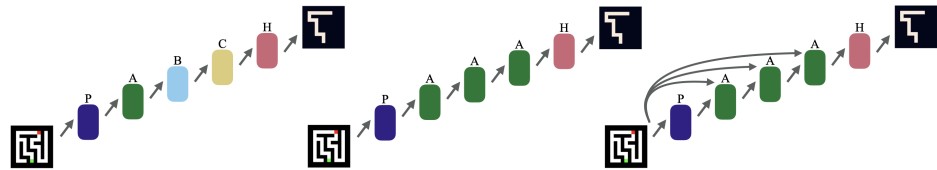

Figure 2: Architecture schematics. Left to right: A feed-forward network, a network containing three recurrent blocks (in green) that share weights, and a recurrent network with recall.

Popular architectures in computer vision typically incorporate skip connections that similarly pass information from earlier layers forward. In fact, empirical evidence suggests that skip connections, for example in highway networks, ResNets, and DenseNets, stabilize training (Srivastava et al., 2015; He et al., 2016; Huang et al., 2017). Our architectural modification is driven by the intuition that a noisy training process creates thinking networks that are imperfect and may leak or distort information over time as features are iteratively fed back through the recurrent unit thousands of times. Recall allows the system to look back at the problem at any time and reproduce any missing or damaged features. To formalize this architectural change, adding *recall* to the network can be expressed using the notation defined above as follows.

$$f_m := h(r_{\text{recall}}^m(p(x), x)), \text{ where } r_{\text{recall}}(\phi, x) := r([\phi, x]) \tag{2}$$

Whereas the input to $r$ at iteration $k$ is usually $\phi_{k-1}$, with recall, the input to $r$ at iteration $k$ is $[\phi_{k-1}, x]$, or the concatenation of the input with the feature map output by the previous recurrence. We add a single convolutional layer to map the input $[\phi_{k-1}, x]$ to a feature map of the same shape as $\phi_{k-1}$. We refer to DT networks with concatenating skip connections as DT-Recall models.

### 3.2 PROMOTING FORWARD PROGRESS THROUGH OPTIMIZATION

We modify the training objective to encourage the system to incrementally make progress from any starting point. We do this by inputting a problem instance and running the recurrent module for some random number of iterations. We then take the output of this process, re-initialize the network with these features as if iteration had just begun, and train the model to produce the correct solution after a random number of iterations.

This incremental training process has two benefits. First, it trains the network to continue improving the quality of partial solutions, even when they contain errors or distortions that creep in from

running many iterations. Second, by choosing features from a random iteration to serve as the initial state for the training step, we prevent the network from internally counting iterations and learning iteration-specific behaviors, such as behaviors that get executed only on iteration five, for example. Rather, the network is forced to learn iteration-agnostic behaviors that are effective at any stage of the problem solving process.

In our implementation, we randomly sample the number of iterations used to generate a partial solution, $n$, and the number of training iterations, $k$, budgeted for the network to improve this partial solution. We then update the network's parameters to minimize loss after $n + k$ total iterations when it starts with the initial partial solution. This is done by detaching the recurrent module's output after $n$ iterations from the computation graph before computing the gradient of the loss at iteration $n + k$. The process above is an analog of truncated backpropagation through time (Jaeger, 2002), with a random starting and end point. During training, we ensure that the sum of $n$ and $k$ is less than a fixed maximum number of iteration $m$, which we call the *training regime*. The incremental loss described above is added to the standard loss computed with a full forward and backward pass through the unrolled $m$-iteration network.

The training loop and computation of the loss are given in Algorithm 1. The incremental progress term is referred to as $\mathcal{L}_{\text{progressive}}$ (or progressive loss), and the contribution to the loss from the fully unrolled network is denoted by $\mathcal{L}_{\text{max\_iters}}$. Note that when the weight $\alpha$ is equal to 1, there is no contribution from the full forward pass and also that $\alpha = 0$ corresponds to full backpropagation through all iterations with no incremental objective.

---

**Algorithm 1** Incremental Progress Training Algorithm

**Input:** parameter vector $\theta$, integer $m$, weight $\alpha$
**for** batch_idx = 1, 2, ... **do**
    Sample inputs $x$ and targets $y$.
    Choose $n \sim U\{0, m-1\}$ and $k \sim U\{1, m-n\}$
    Compute $\phi_n$ with forward pass through $n$ iterations without tracking gradients
    Compute $\hat{y}_{\text{prog}}$ with forward pass of additional $k$ iterations
    Compute $\hat{y}_m$ with new forward pass of $m$ iterations
    Compute $\mathcal{L}_{\text{max\_iters}}$ with $\hat{y}_m$ and $\mathcal{L}_{\text{progressive}}$ with $\hat{y}_{\text{prog}}$.
    Compute $\mathcal{L} = (1 - \alpha) \cdot \mathcal{L}_{\text{max\_iters}} + \alpha \cdot \mathcal{L}_{\text{progressive}}$
    Compute $\nabla_\theta \mathcal{L}$ and update $\theta$
**end for**

---

## 4 EXPERIMENTS

We evaluate our improvements on the benchmark problems available in the Python package `easy-to-hard-data`. The three problems considered are computing prefix sums, finding the optimal path in a two-dimensional maze, and solving chess puzzles. We briefly review the input and output structures for each problem and refer the reader to Schwarzschild et al. (2021a) for more detail, including the data generation process. Note that the architectures we consider are fully convolutional, and produce outputs of the same dimension as their inputs. Furthermore, the training and testing datasets we conside have labels of the same dimension as their inputs. Therefore, a network trained on inputs of one size can then trivially be applied to inputs of a different size.

We begin with the toy problem of computing prefix sums modulo two. The inputs and targets for this problem consist of bit strings. The $i^{\text{th}}$ bit of the target is the mod 2 sum of all bits prior to and including the $i^{\text{th}}$ bit of the input. We control the "difficulty" of the problem by changing the length of the bit string. Note that computing prefix sums of greater length is known to require a greater number of sequential operations (Ladner & Fischer, 1980). All of our training is done on 32-bit strings and we explore the behavior of our models on longer strings, even showing strong performance on 512-bits.

The mazes we consider are two-dimensional square images where the walls are black, the permissible paths are white, and the start and end positions are denoted with red and green squares. The targets for this problem are maps of the same dimension as the input maze, but with ones on the optimal path and zeros elsewhere. We make more challenging datasets by increasing the size of the mazes. All training in our work is done on $9 \times 9$ mazes. Despite this small training size, we benchmark

extrapolation performance on mazes of size $59 \times 59$, and observe models solving mazes as large as $201 \times 201$ with high success rates.

The chess puzzles we consider are mid-game boards represented by twelve $8 \times 8$ planes indicating the positions of all 12 types of pieces. (There are six distinct piece classes and two colors on a chess board.) The goal of this task is to find the best next move, and each target encodes this information in an $8 \times 8$ array with zeros everywhere except the origin and destination of the piece to be moved, which are populated with ones. The chess dataset is sorted by difficulty rating, as determined by Elo scores computed via human trials on Lichess.org. Problems are sorted by difficulty and we train on the first 600K easiest puzzles and test our models on indices ranging from 700K to 1.1M.

In each problem domain, we show that architectures with recall trained with the progressive loss perform best. We include ablation studies lending weight to our claim that both the architecture and the loss modifications are instrumental in boosting logical extrapolation power. For a discussion on hyperparameter choice and training set-ups for all of our experiments, see Appendix A.3. Even with tuned hyperparameters, training is sometimes unstable – e.g. one out of every five maze-solving models fails to train. Code to reproduce all of our experiments is publicly available.[1]

**Prefix Sums** Though Schwarzschild et al. (2021b) only perform tests on strings with up to 48 bits, we use 48-bit and 512-bit testing sets to illustrate the extent of our performance improvements. We train our models on 32-bit strings with a maximum of only $m = 30$ recurrent iterations (indicated in the figures as the shaded 'training regime' section), 400 convolutional filters per layer, and incremental loss weight $\alpha = 1$ when the new loss is used.

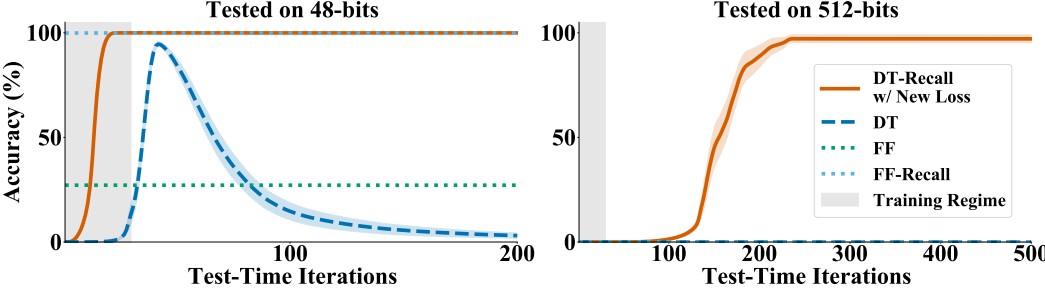

Figure 3: Comparison of our models, DT networks, and feed-forward models on 48-bit (left) and 512-bit (right) data. Our models solve more than 99% and 97% of 48-bit and 512-bit problems, respectively. Note, the curves for prior methods coincide with the x-axis on the right. Shaded regions (in all plots like this) denote $\pm$ one standard error.

In Figure 3, we show that DT networks without our proposed modifications are unable to solve very long binary strings, achieving 0% accuracy on 512-bit data, while DT-Recall models reach 100% after approximately 200 iterations. Notably, our models with recall and progressive loss do not suffer from the sharp decline in accuracy as the number of iterations increases beyond 20. The monotonic increase in accuracy with added iterations in our model is practically useful, as it allows for pre-defining a large iteration number at which to terminate the recurrence rather than having to carefully choose a stopping iteration to maximize performance while avoiding degradation.

We compare our models to a baseline of DT models trained with $m = 30$ and 400 filters per layer, which is both wider and deeper than the models studied by Schwarzschild et al. (2021b). However, to provide a fair comparison, we use the same values for these parameters since we find that all architectures benefit from additional width and depth. We also compare to feed-forwards model without weight sharing of the same effective depth as a 30-iteration DT network.

To better understand the individual effects of our proposed approach, we perform an ablation study. On both test sets, DT-Recall networks trained *without* the progressive loss achieve the same final accuracy as models trained *with* it, but require approximately twice the number of iterations to get there, see Figure 4. Because our proposed objective succeeds in making models iteration agnostic, solutions are often found sooner than DT models that are trained to solve 32-bit problems in 30 iterations specifically. DT models trained with the progressive loss reach approximately 3% higher peak accuracy than DT networks on 48-bit data, which is also sustained for more iterations. However, both these models (without recall) fail on 512-bit data and they suffer from overthinking.

---

[1]Code can be found in the supplementary material.

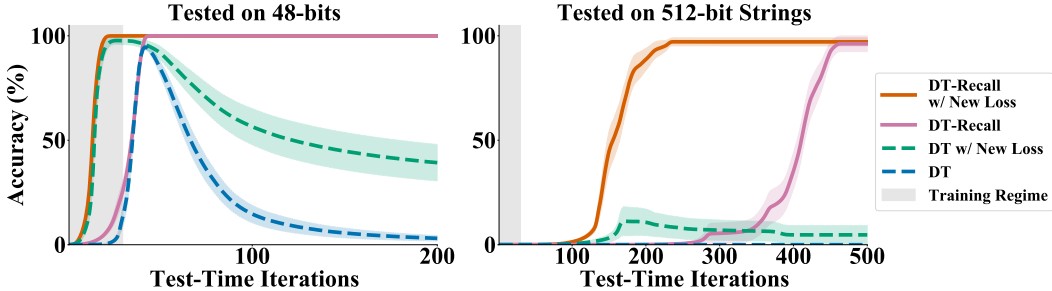

Figure 4: An ablation study showing the value of recall and progressive loss. Our models, which have both, outperform models with neither or only one of these improvements.

**Mazes** As above with the prefix sum problem, we can improve performance on hard mazes by combining incremental training and recall architectures. In particular, while Figure 4 may not convince the reader that our proposed loss is critical since DT-Recall models without it perform very well, with more complex data a drastic difference emerges.

We show in Figure 5 that our models trivially solve the test set of $13 \times 13$ mazes considered to be 'hard' in prior work (Schwarzschild et al., 2021b). Moreover, on the significantly harder test set of $59 \times 59$ examples, our models exhibit strong logical extrapolation, while previous methods, both feed-forward and thinking systems, completely fail. Not only do our models achieve a higher peak accuracy, but they do not overthink, as can be seen by the flat spans in the red curves in Figure 5. In fact, we can push these systems to their limits (and the limits of our hardware) and find that our models can solve nine of ten $201 \times 201$ mazes using more than 2,000 iterations. See Appendix A.9 for an example of an input this large.

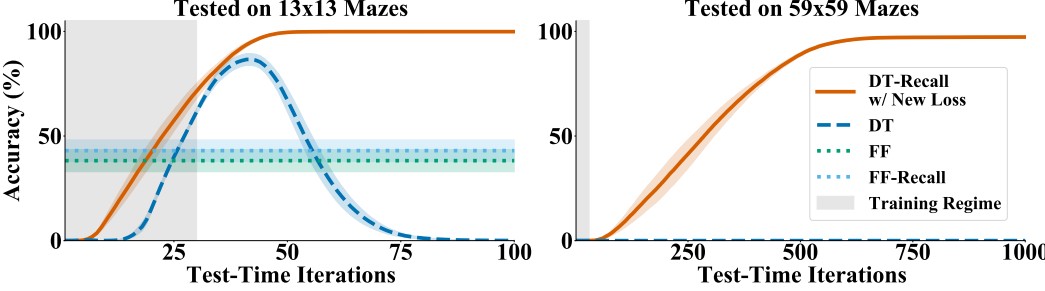

Figure 5: Test accuracy on $13 \times 13$ (left) and $59 \times 59$ (right) mazes as a function of test-time iterations. The horizontal dotted lines correspond to feed-forward models and coincide with the x-axis (0% accuracy) in the plot on the right.

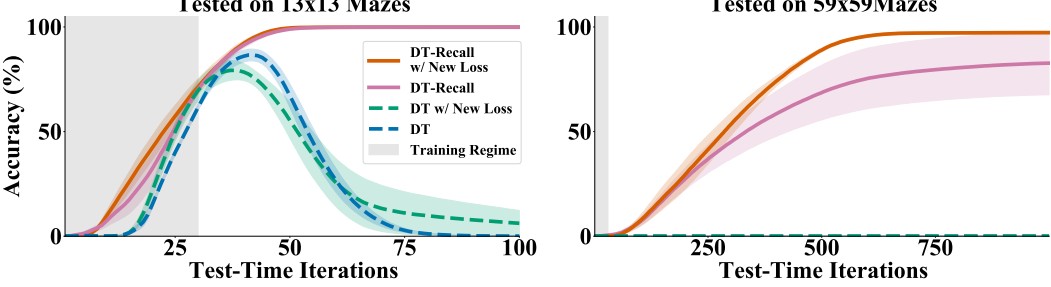

Figure 6: Test accuracy for a variety of models demonstrating the sensitivity to the weight used in the loss computation and to the presence/absence of recall.

We seek to understand the importance of recall and training with the improved optimization procedure. Since the best models in Figure 5 are DT-Recall models trained with 30 maximum iterations and a weighting in the loss of $\alpha = 0.01$, we compare to models with other combinations of design elements. In order to see how critical recall is in overcoming the overthinking problem, we show in the left-hand pane of Figure 6 that without recall all models suffer from overthinking. The decay in each dashed curve and the stability with high iteration counts of the solid curves makes this clear. Notice, however, that testing on $13 \times 13$ mazes does not reveal large differences between DT-Recall

models (with/without progressive loss). For this, we shift our attention to tests on $59 \times 59$ mazes, where DT-Recall models with progressive loss achieve on average 97% accuracy while DT-Recall models without progressive loss only reach an average of 83%. Interestingly, on mazes, we find that the best model has $\alpha = 0.01$ in the loss, a much smaller weight than we use for prefix sums. More on finding the optimal value for $\alpha$ can be found in Appendix A.3.

**Chess Puzzles** We further find that using recall and the progressive loss yields notable improvement on chess puzzles. In Figure 7, these modifications generate a 4% accuracy improvement compared to vanilla DT networks. Moreover, the accuracy of the DT-Recall networks is preserved as the number of iterations increases, while that of the DT networks drops to nearly 0% after 100 iterations. As with the above benchmark datasets, we justify our hyperparameter choices with ablation studies. In the right-hand plot of Figure 7, we show that either removing recall or training without the incremental progress loss will damage performance. Our best models are DT-Recall networks with 512 convolutional filters in each layer trained with a maximum of 30 iterations and a weight for the combination of loss terms of $\alpha = 0.5$. Results from tests on harder puzzles are in Appendix A.4.

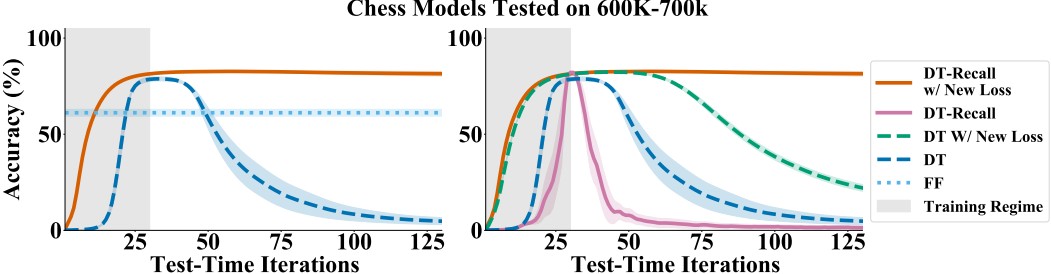

Figure 7: Test accuracy on harder puzzles. Left: Our models compared with baselines. Right: Accuracies of an array of models showing the importance of recall and our loss.

## 5  THE OVERTHINKING PROBLEM

Deep Thinking networks boast impressive capabilities to solve harder problems by thinking deeper, but they are prone to overthinking. In particular, we see in Section 4 that some recurrent networks that can perform logical extrapolation may collapse entirely when performing too many iterations. One observation that can be made from the results above is that the overthinking problem seems to disappear when skip connections are added to provide networks with an uninterrupted view of the input. Another way to describe this is that our models seem to converge to a fixed-point solution as they iterate rather than becoming unstable. This property is desirable and using representative models trained to solve mazes, we explore how robust models are when we manipulate the features during the thinking process. In Appendix A.8, we present similar findings for prefix sum models.

### 5.1  MANIPULATING FEATURE MAPS

First, we investigate sensitivity to adding noise in the feature maps before concatenating the inputs. We examine model behavior when we add Gaussian noise with mean zero and standard deviation one to the features after one iteration of maze solving. Models with recall can still solve $13 \times 13$ mazes even when we perturb the features, but models without recall cannot. See Appendix A.8 for plots, results, and further discussion.

Next, we ask whether or not the initial feature maps (after the projection layer $p$) carry any important information. We test this by replacing the initial feature maps with zeros after 50 iterations. In this case, our DT-recall models naturally regenerate their features and recover to solve the problem again, indicating that the learned algorithm is able to find a solution using the input without the initial projection. See the top-left panel of Figure 20 in Appendix A.8.

### 5.2  MANIPULATING INPUTS

With the notion that our models may embody a convergent process, we turn our attention to investigating how the networks determine when to stop manipulating the representation. Perhaps there is something in the feature maps output by late iterations that tells the network to stop working. We

can test this in two ways. We perturb the input (which we concatenate onto the features) after some number of iterations – first subtly, then by swapping the maze with an entirely different example.

To start, we change the input by moving the end of the maze two steps closer to the start of the maze. We use this new input concatenated with features generated after 50 iterations of solving the original maze. In this case, we see in the bottom-left pane of Figure 20 (appendix) that models with recall correct to solve the new problem in very few iterations.

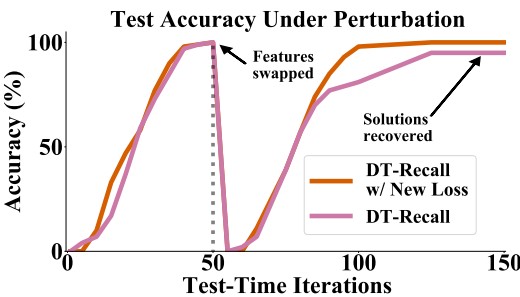

Figure 8: Test accuracy on $13 \times 13$ mazes when features are swapped after 50 iterations.

A final way to test the hypothesis that networks are continually comparing their solution to the problem instance is by partly solving Maze "A," and then swapping the features with those obtained from 50 iterations of trying to solve Maze "B." In other words: if we concatenate the input problem (A) with the features from iteration 50 corresponding to a different maze (B), which maze will the network solve? Clearly, a system without recall will solve maze B. However, with recall, networks will recover and pull the features back to representing a solution for maze A. Figure 8 shows the effect of swapping feature maps.

## 5.3 CONVERGING TO A FIXED POINT

Finally, we can study the convergence by measuring the change in the output at each iteration. A decreasing change in the feature maps with each additional iteration suggests that the network manipulates the representation, moving it closer to one that solves the problem, and that it will hold onto this representation (or stay nearby) once it is reached. Moreover, we seek to qualitatively categorize each model type (architecture/loss pair) as convergent or non-

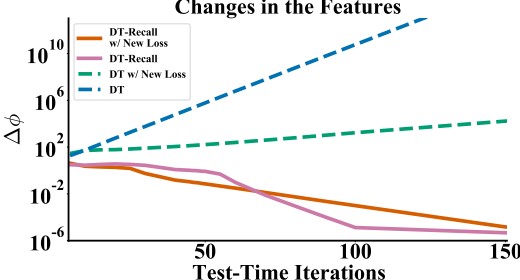

Figure 9: The change in the features when solving $13 \times 13$ mazes. Recall keeps this quantity small.

convergent. To do we measure the change in the solution at each iteration with $\ell_2$-norm of the difference. Figure 9 shows $\Delta\phi(n) := \|\phi_n - \phi_{n-1}\|_{\ell_2}$. We see that our models appear to converge, while DT networks without recall explode, providing another view into overthinking.

## 6 CONCLUSION

In this work, we improve the logical extrapolation power of neural networks. We propose an architecture modification and a new loss that lead to a gigantic generalization leap from easy training data to much more complex testing examples. Furthermore, we show that our models avoid the overthinking trap. We test logical extrapolation with chess puzzles where the spacial dimension is consistent across difficulties and with maze solving and prefix sum computation where our models can extrapolate to larger problems as well. In fact, our models use more than 2,000 iterations, the equivalent of more than 10,000 convolutional layers, to solve the largest mazes we consider.

Learning scalable processes and performing logical extrapolation are difficult tasks for most neural models, but with our architecture and loss, we demonstrate that huge leaps in complexity from training to testing data can be handled without overthinking. Importantly, our neural networks learn end-to-end how to perform logical extrapolation. Observing neural models that handle this important type of domain shift prompts future inquiry into online stopping conditions, through which models will determine on their own when to stop thinking and into applications to even more complex real world problems.

# 7 ETHICS AND REPRODUCIBILITY

The experiments in our work, on academic logical reasoning problems, indicate that our models can extrapolate their reasoning, but real-world domain shift is not limited to logical extrapolation, and our paradigm might fail for numerous unforseen reasons. Thus, we caution against the assumption that our methods will translate to critical applications, which were not tested in this work.

In order to reproduce our findings, refer to the code and the supplementary material. Both publicly available, they provide reference for the exact implementations we use along with specific hyperparameters and training set-ups. The data we use is freely accessible as part of an open-source Python package called `easy-to-hard-data`, and we use version 1.0.0, released in August 2021.

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

## A APPENDIX

### A.1 ARCHITECTURES

The architectures we use in our experiments are very similar to those described by Schwarzschild et al. (2021b), but we provide the details here for convenience.

In all the models, the first layer $p$ is a convolutional layer with $3 \times 3$ filters (or filters of length three in the 1D case) followed by a ReLU non-linearity, which projects the input into feature space. Each filter strides by one, and inputs are padded with one unit in each direction. The number of filters is specified per dataset in Table 1 as the "width" of the network. The internal blocks are standard residual blocks with four convolutional layers (followed by ReLUs) and skip connections every two layers (He et al., 2016). These blocks share weights in recurrent networks and are simply repeated with distinct parameters in feed-forward models. Models with recall have additional convolutional layers that map the concatenated inputs and features $[\phi, x]$ to the shape of the feature maps $\phi$. The feature maps have the same spacial dimension as the input and $w$ channels, where $w$ denotes the width of the model. The final block $h$ is composed of three convolutional layers with decreasing widths (specific numbers are in the Table 1), and ReLUs after the first two. The third and final convolutional layer in $h$ has two channel outputs used for binary pixel classification.

Table 1: Model architecures for all main experiments. Note, we perform ablations where we change the width or the maximum number of iterations and those parameters are indicated where appropriate.

| Dataset | Width | # Channels in $h$ layers |
|---|---|---|
| Prefix Sums | 400 | 400, 200, 2 |
| Mazes | 128 | 32, 8, 2 |
| Chess | 512 | 32, 8, 2 |

### A.2 THE LOSS

Prefix sum problems with $n$ bits are input to the model as a vector $x \in \{0, 1\}^n$, and the output is denoted by $\hat{y} \in \mathbb{R}^{n \times 2}$. The target output is $y \in \{0, 1\}^n$. We compute the loss as follows.

$$\ell(\hat{y}, y) = -\frac{1}{n} \sum_{i=0}^{n-1} \log \frac{e^{\hat{y}[i, y_i]}}{e^{\hat{y}[i,0]} + e^{\hat{y}[i,1]}} \tag{3}$$

where $[\cdot, \cdot]$ indexes the output array. Therefore, for a batch of $B$ instances, the total loss is

$$\mathcal{L} = \frac{1}{B} \sum_{b=0}^{B-1} \ell(\hat{y}_b, y_b). \tag{4}$$

This loss applies to all three problem types we consider. Note that a maze represented by an $m \times m$ input has $m^2$ pixels and the loss can be averaged over those pixels. The same applies to chess, where there are always 64 pixels.

### A.3 HYPERPARAMETERS

We describe the training set-up for each experiment and discuss hyperparameter choices.

Due to instability of training for prefix sum networks, all models are optimized using Adam (Kingma & Ba, 2014) and gradient clipping. Maze models are also trained with Adam, but no clipping is used. Chess models train stably with SGD without gradient clipping. All training is done with a weight decay coefficient of 0.0002 and training with SGD uses a momentum coefficient of 0.9.

In each training run, we hold out 20% of the training data to use as a validation set, and we save for testing the checkpoint with the highest validation accuracy. Additionally, when we compute averages,

we only include models that trained beyond a threshold training accuracy. Our threshold choices for 90% prefix sums and for chess and 99% for mazes.

Coarse experimentation with learning rates and decay schedules is used to determine optimal choices of these hyperparameters for reproducibility and speed of training. Models were trained with exponential warm-up for a small number of epochs before the main training routine. All DT models presented in 4 are trained using early stopping to combat overfitting – network parameters are saved for model with highest accuracy on hold-out set during training. Feed-forward networks are tested with both early stopping and full epoch training, and the best performing class of model is taken as the baseline.

In Table 2, we present the training hyperparameters shared among *all* models presented in Section 4 for each problem instance.

Table 2: Training hyperparameters. Dashes indicate that we did not utiltize those options.

| Task | Optimizer | Learning Rate | Decay Schedule | Decay Factor | Warm-Up Epochs | Epochs | Clip Coefficient |
|---|---|---|---|---|---|---|---|
| Prefix Sums | Adam | 0.001 | [60, 100] | 0.01 | 10 | 150 | 1.0 |
| Mazes | Adam | 0.001 | - | - | 10 | 50 | - |
| Mazes (FF) | Adam | 0.0001 | [175] | 0.1 | 10 | 200 | - |
| Chess Puzzles | SGD | 0.010 | [100, 110] | 0.01 | 3 | 120 | - |

We perform width and depth experiments to demonstrate that our best *prefix sum* models are, roughly speaking, on a plateau with respect to performance on 512-bit data in hyperparameter space. We see from the right panel of Figure 10 that performance increases steadily with depth, but no additional performance is achieved by increasing $m$ past 30. From the left panel of Figure 10, we also see performance steadily increases with network width. We choose to use a width of 400 filters per layer and $m = 30$ because this adequately solves 512-bit data, and it is memory inefficient to use any more for the task at hand.

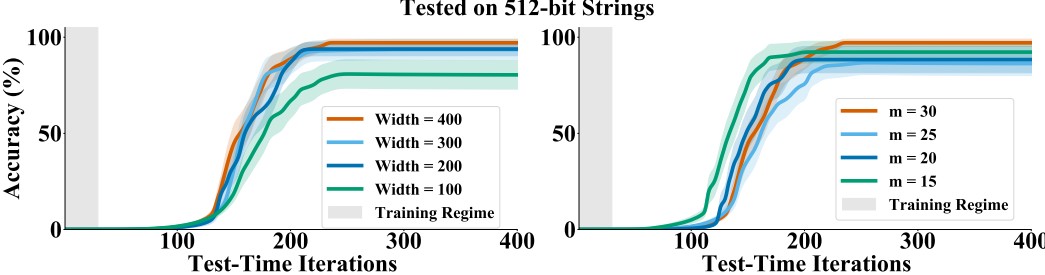

Figure 10: Experiments to determine optimal depth and width for prefix sum models trained with progressive loss ($\alpha = 1$) and recall.

These experiments are performed on prefix sums, which are relatively memory and computationally inexpensive, to exemplify the general correlation between depth, width, and performance observed on all tasks. The choice of depth and width of maze and chess models is also made to be only as large as necessary to avoid excessive memory and computational costs.

We determine the optimal weight to be used it the combination of the loss terms using a coarse grid search. For prefix sums we test models with $\alpha$ values in $[0, 0.25, 0.5, 0.75, 1]$, for mazes we test values $[0, 0.01, 0.1, 0.5, 1]$, and for chess we test values $[0, 0.5, 1]$. We choose the best weight for each task to present in the main experiments.

For evaluating recurrent models, Schwarzschild et al. (2021b) propose a maximum confidence rule. This algorithm takes the output from the iteration at which the outputs have the highest confidence. This is measured by averaging the per pixel confidence, where confidence is computed as the maximum of the softmax over the output channels. In order to fairly compare our recurrent models to theirs, we use the same evaluation technique. For more detail see Appendix A.6.

## A.4 EXTENDED RESULTS

Tables 3, 4, and 5 show the peak accuracy of each of the curves presented in the plots in Section 4.

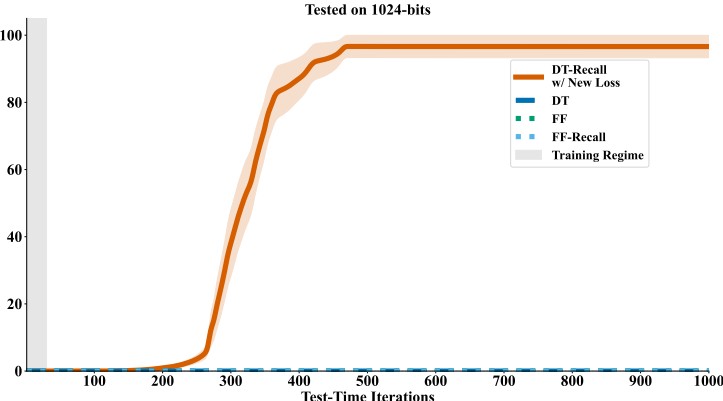

Figure 11: Prefix sum results on 1024-bit inputs.

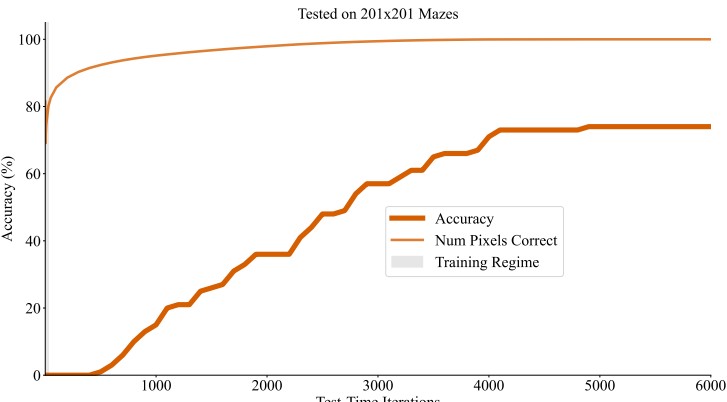

Figure 12: Accuracy curve from a single representative model on $201 \times 201$ mazes. On this plot, we add a curve to indicate what percentage of pixels that are correct to draw attention to the fact that the seemingly low performance of 74% accuracy should be understood in context. Of the 100 mazes in the test set, 20 have one pixel (out of 161,604) wrong. No single maze has more than seven mistakes.

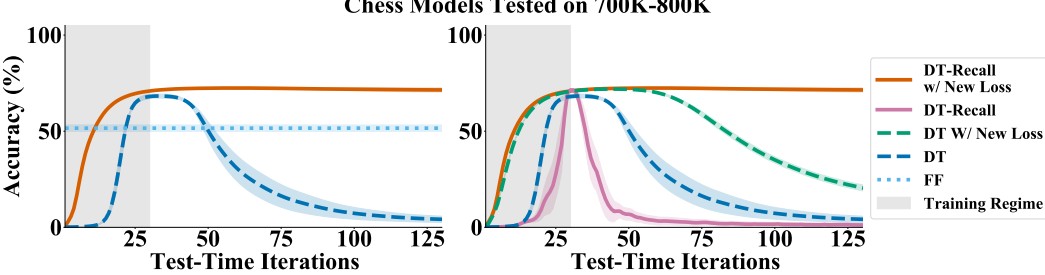

Figure 13: Chess performance when tested on puzzles with indices from 700K to 800K.

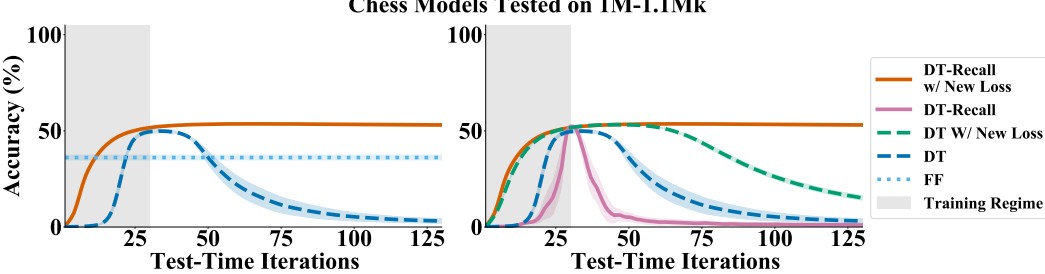

Figure 14: Chess performance when tested on puzzles with indices from 1M to 1.1M.

Table 3: The peak accuracy and corresponding test-time iteration number for prefix sum solving model performance curves in Figures 3 and 4

| Tested on 48-bit Strings | | | | Tested on 512-bit Strings | | | |
|---|---|---|---|---|---|---|---|
| Model | $\alpha$ | Peak Iter. | Peak Acc. (%) | Model | $\alpha$ | Peak Iter. | Peak Acc. (%) |
| DT | 0.0 | 42 | $94.61 \pm 1.19$ | DT | 0.0 | - | $0.00 \pm 0.00$ |
| DT | 1.0 | 27 | $97.73 \pm 1.80$ | DT | 1.0 | 171 | $11.26 \pm 6.90$ |
| DT-Recall | 0.0 | 46 | $99.97 \pm 0.01$ | DT-Recall | 0.0 | 466 | $96.19 \pm 3.73$ |
| DT-Recall | 1.0 | 26 | $99.96 \pm 0.02$ | DT-Recall | 1.0 | 237 | $97.12 \pm 1.88$ |
| FF | 0.0 | 30 | $27.15 \pm 2.56$ | FF | 0.0 | 30 | $0.00 \pm 0.00$ |
| FF-Recall | 1.0 | 30 | $99.87 \pm 0.04$ | FF-Recall | 1.0 | 30 | $0.00 \pm 0.00$ |

Table 4: The peak accuracy and corresponding test-time iteration number for maze solving model performance curves in Figures 5 and 6.

| Tested on $13 \times 13$ Mazes | | | | Tested on $59 \times 59$ Mazes | | | |
|---|---|---|---|---|---|---|---|
| Model | $\alpha$ | Peak Iter. | Peak Acc. (%) | Model | $\alpha$ | Peak Iter. | Peak Acc. (%) |
| DT | 0.00 | 40 | $85.59 \pm 2.81$ | DT | 0.00 | - | $0.00 \pm 0.00$ |
| DT | 0.01 | 38 | $86.08 \pm 3.96$ | DT | 0.01 | - | $0.00 \pm 0.00$ |
| DT-Recall | 0.00 | 94 | $99.94 \pm 0.03$ | DT-Recall | 0.00 | 999 | $82.72 \pm 15.14$ |
| DT-Recall | 0.01 | 70 | $99.88 \pm 0.05$ | DT-Recall | 0.01 | 984 | $97.30 \pm 0.68$ |
| FF | 0.00 | 0 | $38.22 \pm 5.28$ | FF | 0.00 | - | $0.00 \pm 0.00$ |
| FF-Recall | 0.01 | 0 | $43.01 \pm 5.14$ | FF-Recall | 0.01 | - | $0.00 \pm 0.00$ |

Figures 13 and 14 show the test performance of chess models on even harder test sets. It is interesting to note that recall and our loss both still help dramatically, but there is an apparent ceiling on generalizing from the easy puzzles to much harder ones. We leave further investigation into this phenomenon for future work.

## A.5  IN-DISTRIBUTION RESULTS

Information on how models perform in distribution reveals that each class of model fits the training data and generalizes in the classical sense. Specifically, we report the accuracy on a held-out validation set from the same problem difficulty as the training data. For recurrent models, we evaluate the performance using the maximum number of iterations used during training, irrespective of the training objective. See Tables 6, 7, and 8.

Table 5: Peak accuracy and iteration number for chess puzzle performance curves in Figure 7.

| | | Tested on puzzles 600K-700K | |
|---|---|---|---|
| Model | $\alpha$ | Peak Iter. | Peak Acc. (%) |
| DT | 0.0 | 32 | $78.81 \pm 1.04$ |
| DT | 0.5 | 43 | $82.32 \pm 0.10$ |
| DT-Recall | 0.0 | 29 | $82.12 \pm 0.69$ |
| DT-Recall | 0.5 | 57 | $82.69 \pm 0.27$ |
| FF | 0.0 | 30 | $61.17 \pm 1.76$ |

Table 6: Training accuracy and in-distribution validation accuracy for models presented in Figures 3 and 4

| | | Trained on 32-bit Strings | |
|---|---|---|---|
| Model | $\alpha$ | Train Acc. (%) | Val Acc. (%) |
| DT | 0.0 | $99.98 \pm 0.01$ | $100.00 \pm 0.00$ |
| DT | 1.0 | $99.57 \pm 0.38$ | $97.73 \pm 1.80$ |
| DT-Recall | 0.0 | $99.95 \pm 0.02$ | $100.00 \pm 0.00$ |
| DT-Recall | 1.0 | $99.98 \pm 0.01$ | $100.00 \pm 0.00$ |
| FF | 0.0 | $99.76 \pm 0.14$ | $99.76 \pm 0.14$ |
| FF-Recall | 1.0 | $100.00 \pm 0.00$ | $100.00 \pm 0.00$ |

Table 7: Training accuracy and in-distribution validation accuracy for models presented in Figures 5 and 6

| | | Trained on $9 \times 9$ Mazes | |
|---|---|---|---|
| Model | $\alpha$ | Train Acc. (%) | Val Acc. (%) |
| DT | 0.0 | $100.00 \pm 0.00$ | $100.00 \pm 0.00$ |
| DT | 0.01 | $99.98 \pm 0.01$ | $99.98 \pm 0.01$ |
| DT-Recall | 0.0 | $99.91 \pm 0.07$ | $99.92 \pm 0.06$ |
| DT-Recall | 0.01 | $99.77 \pm 0.15$ | $99.74 \pm 0.17$ |
| FF | 0.0 | $99.94 \pm 0.01$ | $99.93 \pm 0.05$ |
| FF-Recall | 0.01 | $99.99 \pm 0.00$ | $99.99 \pm 0.00$ |

Table 8: Training accuracy and in-distribution validation accuracy for models presented in Figures 7

| | | Trained on puzzles 0-600K | |
|---|---|---|---|
| Model | $\alpha$ | Train Acc. (%) | Val Acc. (%) |
| DT | 0.0 | $99.98 \pm 0.02$ | $92.94 \pm 0.34$ |
| DT | 0.5 | $99.90 \pm 0.01$ | $94.16 \pm 0.02$ |
| DT-Recall | 0.0 | $99.77 \pm 0.02$ | $100.00 \pm 0.00$ |
| DT-Recall | 0.5 | $99.34 \pm 0.02$ | $94.18 \pm 0.07$ |
| FF | 0.0 | $96.89 \pm 0.58$ | $83.24 \pm 1.22$ |

## A.6 EXIT RULES

When a DT network processes a single input, it can produce output at each iteration. When we choose a number of test-time iterations $M$, we are specifying the maximum number of iterations, but we may choose from any of those to extract a single output. A naive approach is to take that output at iteration $M$; we call this We call this the *default exit rule*. A slightly more complicated process, and what Schwarzschild et al. (2021b) do, is to take the output from iteration $m^*$ which solves the following maximization problem. Let $\hat{y}^{(m)}$ be the output from iteration $m$, and let $\hat{y}^{(m)}[k, j]$ be the $(k, j)$ entry in the output that corresponds to the confidence that the $k^{\text{th}}$ pixel is in class $j$, finally let $k \in \{1, 2, ..., K\}$ and let $j \in \{0, 1\}$.

$$m^* = \arg\max_m \frac{1}{K} \sum_k \max_j (\hat{y}^{(m)}[k, j]) \tag{5}$$

Choosing the output using this maximization technique is called the *maximum confidence exit rule*.

In order to stay faithful to the prior work, all plots in the main body of this paper use the *maximum confidence exit rule* (Schwarzschild et al., 2021b). In this section, we compute accuracy at iteration $n$ by using the output after $n$ iterations instead of taking the highest confidence output at all iterations up to $n$. We call this the *default exit rule*.

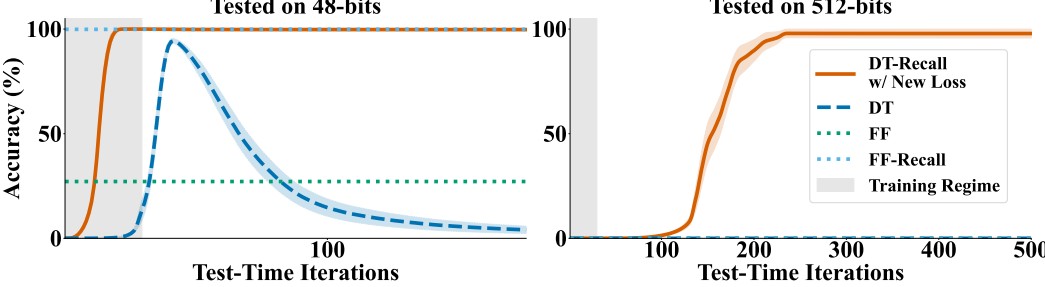

Figure 15: Test accuracy on 48-bit (left) and 512-bit (right) prefix sums as a function of test-time iterations **using the output at each iteration**. The horizontal dotted lines correspond to feed-forward models and coincide with the x-axis (0% accuracy) in the plot on the right. Shaded regions denote $\pm$ one standard error.

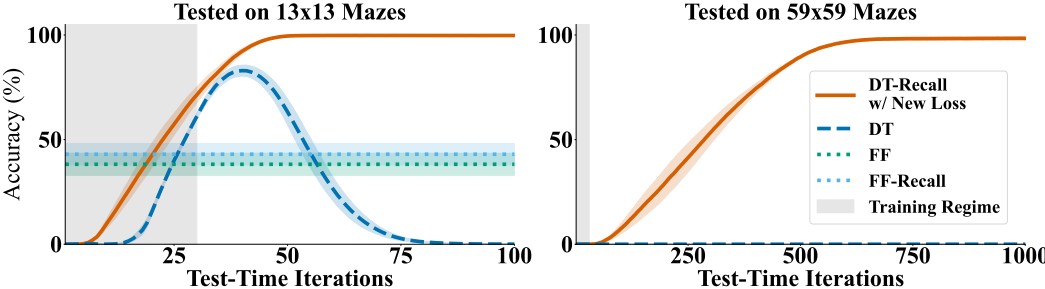

Figure 16: Test accuracy on $13 \times 13$ (left) and $59 \times 59$ (right) mazes as a function of test-time iterations **using the output at each iteration**. The horizontal dotted lines correspond to feed-forward models and coincide with the x-axis (0% accuracy) in the plot on the right. Shaded regions denote $\pm$ one standard error.

To better understand the evolution of confidence as the iteration count increases, we plot this in Figure 18. Note the confidences are high in the early iterations even before the models are correct.

## A.7 HARD TO EASY

One natural query about our models is how well they perform on test sets that are easier/smaller than the data used for training. Figure 19 shows that recurrent models can generally solve easier/smaller problems in fewer iterations. Even though they solve the easier problems in fewer than the maximum number of iterations used in training, we still see that models without recall suffer from overthinking.

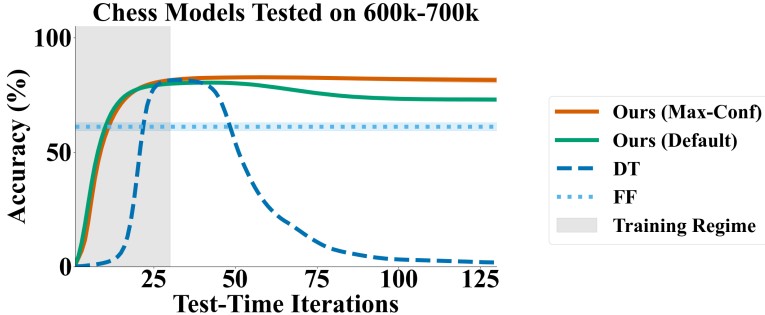

Figure 17: Test accuracy on chess puzzles with indices 600K to 700K as a function of test-time iterations. Exit

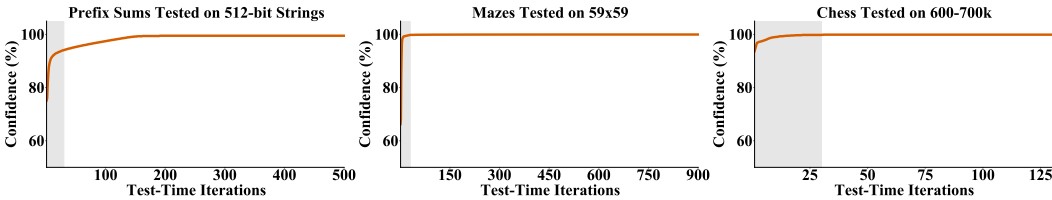

Figure 18: Evolution of confidence of representative models for each dataset as iteration count increases. Left: prefix sums, center: mazes, right: chess.

### A.8    MORE ON THE OVERTHINKING PROBLEM

The experiments we present in Section 5 shed light on how some models overcome the overthinking problem. Here, we present additional plots and tables to give a more complete picture of our findings. Table 9 shows the results from several experiments. The first column indicates the average number of iterations to solve a maze and the best accuracy (over all the test iterations considered). Then, we examine the behavior when the features are perturbed after the first iteration by adding mean zero and variance one Gaussian noise ('Noise' column) and we find that this completely destroys the models without recall while only slightly slowing down those with recall. Next, we see the same trend when we replace the features at the first iteration with arrays of zeros ('Zeros' column). Finally, we swap the features after 50 iterations with those obtained from solving an entirely different maze, and again the models without recall cannot recover from this, but those with recall achieve near perfect accuracy. Figure 20 shows accuracy curves for these experiments. The bottom right plot in Figure 20 shows the norm of the difference in features when we input random noise instead of actual mazes. This plot shows that the stable behavior of our models is not limited to the portion of input space occupied by real mazes, rather the process they learn exhibits convergent behavior even on random inputs.

Table 9: Average iterations to solution, with the peak accuracy in parentheses. We evaluate maze solving models on $13 \times 13$ mazes, where we intervene in the solving process in different ways.

| Model | Clean | Noise | Zeros | Swapped |
|---|---|---|---|---|
| DT | 27.68 (87 %) | - (0%) | - (0%) | - (0%) |
| DT w/ New Loss | 16.87 (95 %) | - (0%) | - (0%) | - (0%) |
| DT-Recall | 23.54 (100 %) | 35.70(100 %) | 35.04 (96 %) | 33.92 (97 %) |
| DT-Recall w/ New Loss | 22.10 (100 %) | 25.10(100 %) | 22.48 (100 %) | 28.79 (100 %) |

Similar experiments on prefix sum computation reveal that recurrent models with recall seem to converge nicely as well. Note, the top two plots in Figure 21 come from identical experiments as those defined above. Whereas with mazes we shortened the optimal path, in this setting we flip single bits in the input string. We explore the response of the model to flipping bits at different indices after 50 iterations (when the models have already solved the initial 48-bit problem). In the lower left plot in Figure 21, we see that the recovery time as a function of which bit is flipped reveals two things: (i)

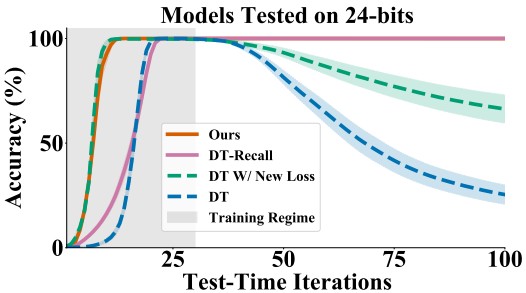

Figure 19: Accuracy as a function of iteration for prefix sum models when generalizing from hard (32-bit strings) to easy data (24-bit strings).

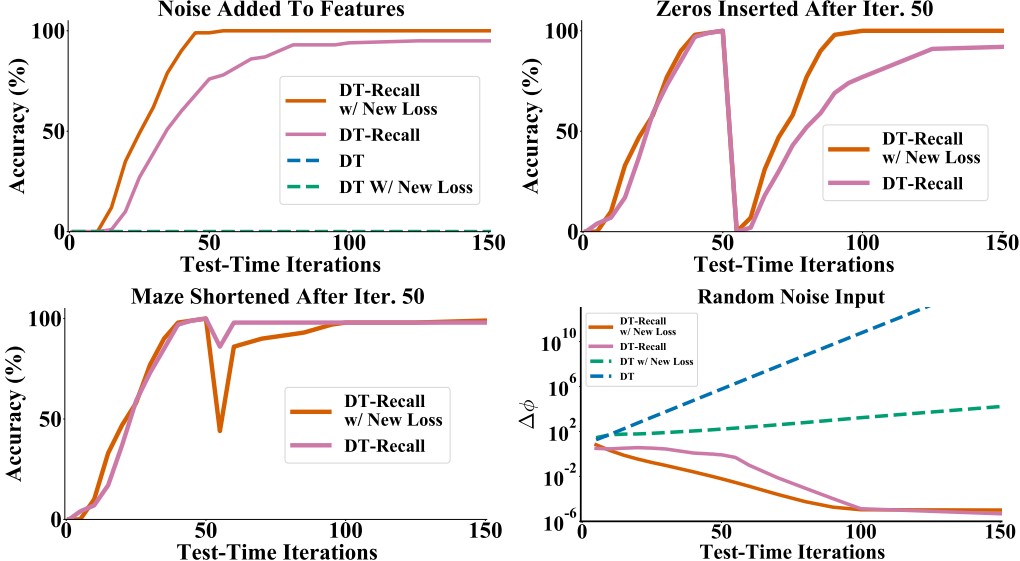

Figure 20: on 13 x 13 mazes.

our models can recover from single bit flips, and (ii) the recovery time decays linearly with the index of the bit flipped. Higher indices are closer to the end and affect fewer bits in the output than lower ones.

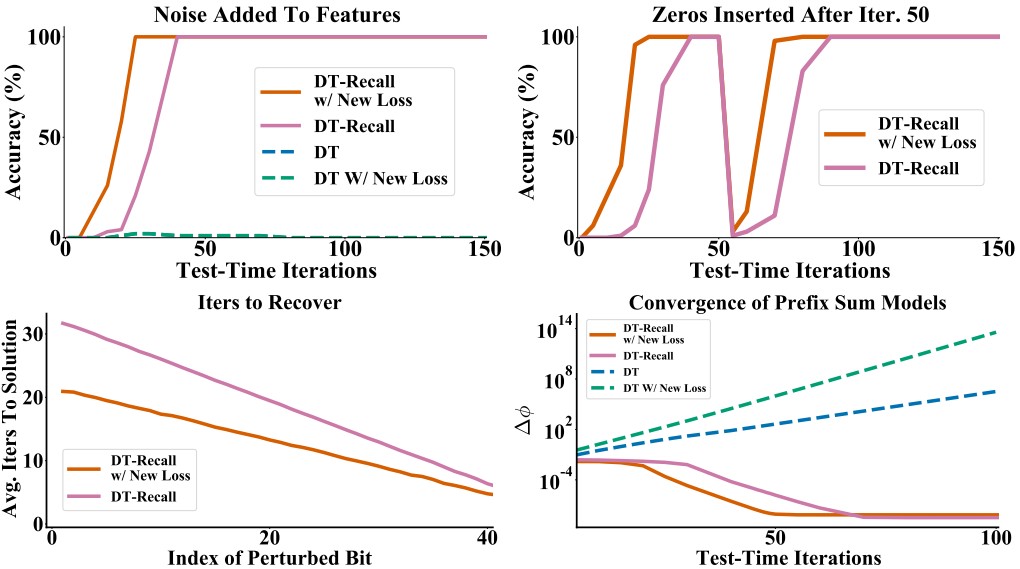

Figure 21: on 13 x 13 mazes.

## A.9   A FUN MAZE

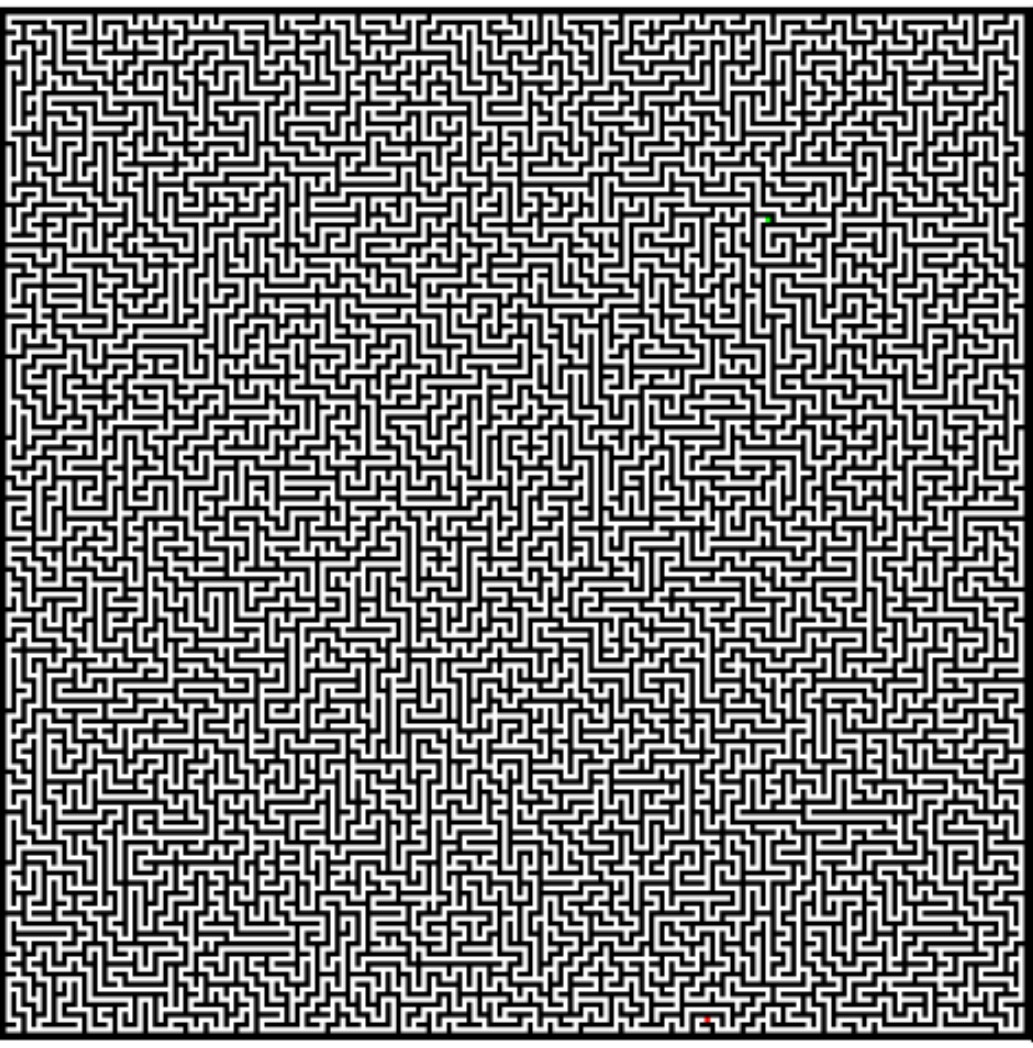

Figure 22: Fun for the whole family! We leave solving this $201 \times 201$ maze as an exercise for the reader. The green starting dot is near the center of the upper right quadrant. The red ending point is on the second row from the bottom.

