# OpenReview forum: "Thinking Deeper With Recurrent Networks: Logical Extrapolation Without Overthinking"
_ICLR.cc/2022/Conference — ICLR 2022 Submitted_

### Official Review · Reviewer_ue13 · 2021-10-25

**Correctness:** 3
**Technical Novelty And Significance:** 2
**Empirical Novelty And Significance:** 2
**Recommendation:** 5
**Confidence:** 4

**Main Review:**

Strengths:
 - The paper clearly states the problem, and convincingly shows a solution.
 - The paper is pretty well written

Weakness
 - The paper is evaluated on a very new dataset, which seems ideal for the kind of method suggested, and the only real comparison is the work they directly build on. You can always find a dataset for which your method is the best, and this paper has a bit of that feeling. Except for the chess positions the examples are new and toy-like. Why do we need these new benchmarks? Why not evaluate on some more standard benchmarks, e.g. image classification/segmentation, simple QA (bAbI for instance), etc.
Also the bit sums and maze path finding are trivially solved with standard hand-coded algorithms. Why would you want to use a neural net on them? If the benchmarks are not interesting problems in their own right, they might still be nice benchmarks because they nicely exemplify some specific problematic environments, but then it must be very clearly explained how this ties back into (better) solving real-world problems. The chess positions task is the nicest example problem, but then again, how does this compare to something like AlphaZero, or any other modern MCTS approach? What's the benefit? Computational? Generalization? This must be explicitly explained.

 - The paper doesn't discuss and compare to the very relevant work "Recurrent Relational Networks" (RRN) [1]. RRNs are recurrent graph neural networks, and very very similar. Both neworks have the same output(recurrent_with_recall(embed(x))) structure and the RRN measures a loss on every step of recurrent computation similar to the modified loss. The RRN also learns a convergent algorithm and generalizes to harder tasks. I would very much appreciate a comparison to a RRN network in all of the new benchmarks, or conversely evaluating on some of the same benchmarks as in [1]. I suspect the performance will be very similar for the two networks.

 - The loss weight alpha hyperparameter is kind of inelegant. Have the authors tried simply measuring a loss on every recurrent iteration like in [1]? This would also encourage a convergent algorithm, and would be simpler.

[1] - Palm, R. B., Paquet, U., & Winther, O. (2018). Recurrent Relational Networks. In 32nd Conference on Neural Information Processing Systems (NeurIPS 2018), Montréal, Canada (Vol. 31, pp. 3368-2278).

**Summary Of The Paper:**

The paper proposes two modifications to recurrent neural networks that enable them to extrapolate to larger problems than seen during training.

The problems (generalizations) are
1) pathfinding in a maze (larger mazes),
2) binary prefix sums (larger bit strings)
3) Evaluate the best chess move given a position (harder positions).

The modifications are:

1) Add the initial problem features to every step of the recurrent computation, coined "recall" in the paper.
2) train on a combination of a regular loss with m recurrent iterations and with a loss with m=n+k iterations where the gradient is not tracked for the first n iterations.

The paper shows that the modified recurrent network learns an algorithm that converges to the correct result with more recurrent iterations, thus overcoming the problem of "overthinking" (divergence really) as coined in earlier papers.

The paper further shows that the learned convergent algorithm "extrapolates" to larger/harder problems. For instance, the networks are trained on 32-bit strings, and evaluated on 512-bit strings, and trained on 9x9 mazes and evaluated on up to 201x201 mazes.

The paper compares their method, ablations w.o. "recall" and modified loss, MLPs and MLPs with "recall", and show that only their proposed network learns a convergent solution that extrapolates to the larger/harder problem instances.

**Summary Of The Review:**

Decent paper, but should be evaluated on more common benchmarks / less toy settings and should discuss and compare to [1].

---

> ### Author Response · Authors · 2021-11-22
> **Response to Reviewer ue13**
>
> We agree that careful consideration of datasets is critical in producing quality research in this field. On that note, we point out that prefix sum computation is not quite a toy problem, rather it is one that has a lot of attention (and real-world use) in the algorithms space [2, 3, 4].  Additionally, chess is known to be a difficult problem, and learning to extrapolate from labelled data (rather than RL) could be a cost effective, interesting, and realistic approach. Lastly, we note that mazes provide a totally different, albeit very applicable, search problem. Path finding has far reaching applications, and we train networks to perform scalable algorithms that generalize from a set of tiny examples to instances several orders of magnitude larger. With regard to the well known benchmarks you mention: We are studying logical extrapolation, and in commonly used well curated datasets for tasks like image classification, there is often no clear delineation of easy and hard examples. For these reasons, we feel that sticking with the three problems used in the work that we build on is the best choice.
>
> Thank you very much for bringing [1] to our attention. This work is interesting and clearly relevant.  We want to point out that [1] states that, in contrast to their graph neural network approach, multi-layer architectures which take in the entire problem simultaneously “as an input and output the entire solution in a single forward pass” are ineffective at solving relational reasoning tasks (of which we consider chess to be a clear example).  We further note that Palm et al. require mapping all problem types to graph representations whereas our recurrent module can operate on the image-format inputs as they are. Nonetheless, we agree that comparisons of the two models on the same benchmark task would be interesting, so we are running these experiments now, and we will update the draft to include results as soon as possible.
>
> Lastly, we have tried training by measuring the loss at every iteration, and we find that our models don’t train as well on the tasks we consider, and this procedure also requires significantly more computation.
>
> [1] - Palm, R. B., Paquet, U., & Winther, O. (2018). Recurrent Relational Networks. In 32nd Conference on Neural Information Processing Systems (NeurIPS 2018), Montréal, Canada (Vol. 31, pp. 3368-2278).
>
> [2] Bahig, Hazem M., and Khaled A. Fathy. "An efficient parallel strategy for high-cost prefix operation." The Journal of Supercomputing 77.6 (2021): 5267-5288
>
> [3] Blelloch, Guy, et al. "SPAA'21 Panel Paper: Architecture-Friendly Algorithms versus Algorithm-Friendly Architectures." Proceedings of the 33rd ACM Symposium on Parallelism in Algorithms and Architectures. 2021.
>
> [4] Cole, Richard, and Uzi Vishkin. "Faster optimal parallel prefix sums and list ranking." Information and computation 81.3 (1989): 334-352.

---

> > ### Comment · Reviewer_ue13 · 2021-11-24
> > **Response**
> >
> > Thank you for the rebuttal.
> >
> > w.r.t. [1]
> >  - A grid is an instance of a graph, so [1] is a more general approach, not the other way around. That's still fine. Your method might still be better or at least better on grids. But it's not true that [1] cannot handle grids, or is somehow less general.
> >  - Your method and [1] both claim that in order to "reason"/"logical thinking" you need a recurrent, iterated computation (with recall). Your paper does not contradict claims in [1], they support them in my opinion. The fact that your method (and [1] for that matter) can be unrolled into a single deep forward computation, does not mean that they are not recurrent.
> >
> > Thank you for doing the experiment with a loss on every step.

---

> > > ### Author Response · Authors · 2021-11-24
> > > **Follow up**
> > >
> > > Thanks for your points regarding [1].  We did not mean to imply that our models are more general than the GNN approach, just that they ingest data in different formats, as we agree that GNNs are incredibly general and CNNs perform a special case of message passing.  Moreover, our interpretation of the statement from [1] was that relational networks are essential, but we see now that another completely plausible interpretation is that recurrence is essential, and we appreciate your perspective.

---

### Official Review · Reviewer_ajNt · 2021-11-01

**Correctness:** 3
**Technical Novelty And Significance:** 2
**Empirical Novelty And Significance:** 1
**Recommendation:** 5
**Confidence:** 4

**Main Review:**

Pros:
- The paper gives solid extensions to the idea of deep thing networks (Schwarzschild et al., 2021). The writing is generally clear and easy to follow. The modifications are well-motivated,  addressing the weaknesses of the original model.
- The experiments demonstrate well the effectiveness of the modifications, outperforming the baselines by a significant margin. There are detailed ablation studies and behaviour analyses that give insights into the inner working of the model.
Cons:
- The modifications are incremental, using common techniques like residual connection and truncated backpropagation through time. Unsurprisingly, using these techniques will help improve the performance.
- The tasks are toyish. It would be more persuasive if the method could work for realistic data or some classical benchmarks.
- The main manuscript misses some details of the method (see questions below).

Overall, I like the idea of this paper, which proposes an interesting way to exhibit logical generalization by allowing recurrent networks to "think" more during inference. However, as mentioned above, compared to the original work, the proposed modifications are marginal. It is straightforward that using residual connections is critical to training deep networks (to combat missing input information or gradient vanishing, etc.).  The progressive loss is more interesting, yet it is unclear from the writing that this is a novel contribution to the training of RNNs (e.g., compared to other auxiliary training for RNN as in Trinh et al., (2018), what is the difference and advantage of using the proposed method).

Regarding experimental results, only Chess Puzzle data seems real. But here, the result is not impressive as there is only a 4% improvement in terms of peak accuracy. The performance of the proposed method on the other two synthetic data is promising.  I am curious whether the method only works for these specific tasks or is extendable to other problems: sequential inputs such as Copy, Associative Recall, graph reasoning (Graves et al., 2016) or natural data (text, images).

Questions and comments:

- It is reasonable to use CNN architecture for image-like inputs. However, for prefix-sum, the input is a sequence of 0 and 1. Did you apply CNN here?
- What is the loss function? To make the paper self-contained, in Sec. 3 or 4, you should describe the output format and the loss used to train the model (from Appendix A, it seems that the final output is a 2d bitmap?).
- The procedure of selecting the inference iteration for performance measurement should be mentioned in the main manuscript.
- The analysis would be more informative if the authors could compare the results of other iterations with that of the peak one (how much difference? Did they converge to the final output as suggested in Fig. 9?)
- For prefix-sum, did you test with longer sequences? Can your method generalize to 1024-bit? Why there is no performance visualization for maze 201x201?
- In the paper, the feed-forward model is a weak baseline. The experiments could be stronger with advanced architectures such as Transformer or self-attention model.
- Fig. 8, More explanation on why the new loss helped more in the setting will be appreciated.
- Fig. 9, the y-axis values go up to 10^10. How's that possible? How does the new loss help in this case?

**Summary Of The Paper:**

The paper proposes two modifications to recurrent neural networks that help improve generalization on three synthetic tasks. In particular, the authors implement a recall mechanism through residual connections to prevent the recurrent network from losing original input information after many iterations. In addition, they randomly apply truncated backpropagation through time to the computation graph to build an auxiliary loss, which diminishes overfitting to the number of recurrent steps. The trained network can solve significantly harder problems by performing more recurrent iterations, showing strong generalization ability.


**Summary Of The Review:**

Overall, I like the idea of this paper, However, compared to the original work, the proposed modifications are marginally significant.

---

> ### Author Response · Authors · 2021-11-22
> **Response to Reviewer ajNt**
>
> We appreciate the thorough feedback.  We agree that recall/skip connections are known to help train DNNs, and we think it is interesting to point out that in our case, the recall connections do not mitigate vanishing gradients. Additionally, we choose to follow the work we most directly build on and use the datasets used there to highlight how much of an improvement we see even on the problems that DT nets were originally designed for. We agree that applications to computer vision and NLP are interesting and deserve careful attention in future projects.
>
> **The loss function:** Thank you for bringing this to our attention. We use the cross-entropy loss for all the problems. Specifically, we output 2 logits for each pixel in the input and average the cross-entropy loss corresponding to the two class classification problem at each location. This has now been written out formally in the appendix.
>
> **Inference iteration:** Thank you for suggesting that we clarify this. The maximum confidence exit rule and the default exit rule are now both described in detail in Appendix A.6.
>
> **Comparison across iterations:** Yes, for models where the features converge, we see the output converge as well. The outputs at each iteration can be compared using the plots we added to Appendix A.6 on exit rules. Specifically, performance using the default exit rule should provide the insight you ask about.
>
> **Even harder test data:** We have now tested our models in the settings you mentioned. They generalize well to 1024-bit prefix sums and to 201x201 mazes, and these results have now been added to the appendix.
>
> **The new loss and Figure 9:** The values grow (roughly exponentially) in this figure indicating that each iteration (residual block) roughly increases the norm of the features multiplicatively for some models. One possibility is that the new loss mitigates this blow up since it keeps the network from learning iteration specific behaviors. This keeps the features from landing in a sweet spot on *only* one iteration.
>
> **Baselines:** Thank you for the input on baselines! Our aim is to boost the performance of DT networks. The true baseline in our work is actually the previous DT networks. While there are algorithms and architectures that are best suited for specific problems, like self attention for sequences or depth-first-search for mazes, we are most interested in showing how much better our DT nets do compared with earlier versions.
>
> **CNNs for prefix sums:** Yes, we used CNNs for the prefix sum as well. The architecture has 1D convolutions and the results indicate that this is an appropriate choice.

---

### Official Review · Reviewer_Ap3D · 2021-11-02

**Correctness:** 3
**Technical Novelty And Significance:** 3
**Empirical Novelty And Significance:** 3
**Recommendation:** 6
**Confidence:** 4

**Main Review:**

Both the underlying approach, and the proposed extensions, are compelling, and the results are promising. It is good to see results across a range of task domains. The analyses on convergence and overthinking in the latter part of the paper are also very nice. I think there are a few issues that need to be sorted out, but am happy to raise my score if the authors can address these concerns:

- The only baseline considered is a feedforward version of the primary model (i.e. in which parameters are not shared across iterations). This seems like a good comparison, but other baselines should be considered as well. How important is the particular recurrent architecture employed here, in which the output is fed back into the model as input? How might an LSTM, with a recurrent hidden state, or a model with an external memory, perform on the generalization benchmarks that are considered in this paper?
- The proposed 'progressive loss' seems to be useful primarily for getting these kinds of models to solve the task in the shortest number of iterations of possible. The reason seems to be that a model trained only on a fixed number of iterations has no reason to arrive at the correct answer any sooner than the point at which a loss will be computed. Given that, I wonder whether the proposed method is the simplest or best way to accomplish this. Would it work to simply train models on a randomly sampled number of iterations, or to penalize longer processing (as is done in 'adaptive computation time')? It would be good to compare the proposed approach to these alternatives.
- In the appendix, the authors state that 'when we compute averages, we only include models that trained beyond a threshold training accuracy.' Does this mean that the reported results are only for a subset of models that reached some training criterion? How many models failed to reach this criterion, and are the results qualitatively different when no such criterion is used?
- Models are evaluated by taking the iteration with the highest confidence, rather than simply using the final iteration. Do the models still perform just as well at generalization to more complex problems if the final iteration is used?
- On the prefix sum and maze problems, the 'recall' element seems to be useful primarily for preventing the 'overthinking' phenomenon, and, for those task domains, this element of their proposal appears to be highly effective (as it prevents overthinking even without the 'progressive loss'). However, on the chess problems, the version of the model with recall, but without the 'progressive loss', still suffers from overthinking. What might explain this discrepancy?

Minor issues and additional questions:

- How does confidence evolve over time? Do the models generally become more confident with a greater number of iterations? If so, could confidence be used as a signal for autonomously selecting how many iterations to perform, rather than having to select this by hand?
- When reading the paper for the first time, it is unclear where the results in Figure 1 come from. I'm assuming these results employ both of the proposed extensions, but it would be helpful to specify that this is the case (i.e. that 'vanilla' thinking systems would not be capable of producing these results).
- The legend for table 1 of the appendix says 'perfrom' instead of 'perform'.
- The legend for table 4 in the appendix references the prefix sum task, but the table appears to contain results from the maze task.
- At the end of the main section with results on the maze task, the reader is directed to section A.3 to see an example of a 201x201 maze, but this is actually in section A.7.




**Summary Of The Paper:**

This paper proposes two extensions to the recently proposed recurrent 'thinking systems', in order to enable them to better generalize from training on simple problems to testing on more complex problems. The proposed extensions involve 1) giving the model access to a cure indicating the to-be-solved problem at each time step ('recall') and 2) a method intended to prevent the model from learning behaviors specific to particular iterations (so as to enable generalization to more complex problems via a larger number of iterations). When combined, these extensions enable generalization to significantly more complex problem instances across three separate task domains.

**Summary Of The Review:**

The proposed approach is compelling, and the results are promising, but there are a few issues that needed to be sorted out, including some additional baselines, and clarification of the selection criteria for the reported results.

Update after discussion period: the authors included a number of supplementary results and informative additional control experiments. I still think that the results would be more convincing if compared to a broader range of competitive baselines, but I think these new results/experiments are a significant enough improvement to merit a score increase.

---

> ### Author Response · Authors · 2021-11-19
> **To Reviewer Ap3D**
>
> Thank you for the detailed and thorough review and the positive feedback! We address each of your points below.
>
> **On the progressive loss:** Training models with a random number of iterations is a special case of our proposed loss, where we set n=0 for every batch. This is an interesting point to look at, and In fact, we have now trained models like this. They perform well, but no better than when n is randomly sampled, and it takes more computation. By using the progressive loss we not only encourage iteration agnostic behavior, but we save computational cost as we need to backpropagate through fewer iterations on average. We agree it would be fascinating to compare a version of our models that are penalized for run time to models trained with ACT, developing that is outside the scope of this project.
>
> **On training convergence:** For prefix sums, 6 of the 20 models with progressive loss and no recall in Figure 4 were filtered this way. All 20 models in the other categories were unaffected by the filtering criterion. When training on mazes, we find that one out of every five models when trained with recall does not meet our criteria. Finally, about half of the models trained on chess do not converge and our results are averages over several convergent training runs for each model. There is no qualitative difference in the comparisons (and thus our claims) when including non-convergent models, but overall the average performances are lower (as expected).  We do want to stress that we do not require any testing data to select the models.
>
> **Which iteration to use:** When we use the final iteration (what we call the `default exit rule` we see almost the same exact results from our models. The previous DT nets do slightly worse. For that reason, we stayed faithful to the work we build on, leading to fair comparisons. We have now added plots with those curves to the appendix.
>
> **Chess** is a problem that has proven to be difficult from a computations standpoint. The easy-to-hard leap in the data may be characteristically different.
>
> The main contributions of our work are improvements to previously developed DT nets, and we compare to the non-weight-shared versions in order to further highlight the value of weight sharing. Algorithms that can flawlessly solve some of the problems we consider exist, and our methods are not state-of-the-art from that perspective.  Problem specific approaches, like LSTMs for sequences or MCTS for chess, certainly may perform better. It is our goal to study non-problem specific architectures and explore logical extrapolation in several domains.
>
> **To address the minor points:**
>
> We do use the confidence as a signal for which iteration to use (maximum confidence rule proposed in prior work), but we leave for future work development and analysis of autonomous stopping methods. We have addressed the other minor issues in the updated draft.

---

> > ### Comment · Reviewer_Ap3D · 2021-11-19
> > **Reply to authors**
> >
> > Thanks to the authors for this reply. Here are my thoughts:
> >
> > **Progressive loss:** I don't think it's correct that the alternative method I'm proposing requires more computation, and in fact I think it actually requires less computation. In the proposed method, $n$ forward passes are first applied, where $n$ is sampled from a uniform distribution between $1$ and $m$. Then, this is followed by $k$ additional forward passes, where $k$ is sampled from a uniform distribution between $1$ and $m-n$. Finally, gradients are computed by backpropagating through the final $k$ iterations, but not the initial $n$ iterations. It is true that $k$ will tend to be larger if $n$ is always set to $0$, since the upper bound on the distribution for $k$ depends on $n$. But the key question is whether it's really necessary to perform the initial $n$ forward passes, or whether the method works just as well when using only the final $k$ forward and backward passes. One could just as easily sample $k$ according to the exact scheme described in the paper (or sample $k$ from a distribution with a smaller range), and then simply omit the $n$ forward passes altogether, which would therefore reduce computation. If that worked just as well, that would be a preferable method since it is both simpler and requires less computation. If that didn't work just as well, and it is for some reason important to the method to perform the $n$ initial forward passes, then including the simpler method as a baseline would be useful for justifying the more complex method.
> >
> > I also don't agree that it is outside of the scope of the current work to compare to a model that is penalized based on the number of time steps used. The primary purpose of the progressive loss seems to be to get the model to arrive at a solution in the smallest number of time steps possible, while still allowing for the possibility of the extended computation on more challenging problems. A natural alternative, employed by previous work (e.g. ACT) is to allow the model to determine how many iterations to perform, but attach a cost to additional iterations. In the 'related work' section, there's some suggestion that maybe these methods won't work well in the kind of OOD generalization regimes studied here, which seems plausible, but that point would be far more convincing if it was accompanied by an empirical evaluation.
> >
> > **Training convergence**: I think this point needs to be explicitly mentioned in the main text, as opposed to only in the Appendix. Specifically, it should be mentioned that 1 out of 5 models fail to train on the maze problems, with a pointer to some figures in the Appendix showing results without using this filtering criterion.
> >
> > **Default exit rule**: It is reassuring to see that the results for prefix sums and mazes look very similar when using the default exit rule. Thanks to the authors for including those results. Do the results also look similar when using the default exit rule on the chess problems?
> >
> > **Confidence over time**: I think it's reasonable to leave the question of autonomous stopping to future work, but I'm still curious to see how confidence evolves over time, wouldn't this be straightforward to plot? If confidence did indeed generally increase over time, that would suggest an obvious approach for future work.

---

> > > ### Author Response · Authors · 2021-11-22
> > > **Follow Up**
> > >
> > > We have now run experiments where we penalize the run time by weighting the loss with increasing coefficients as a function of iteration. Models trained this way exhibit worse generalization to test sets which are harder than their training data. Specifically, on prefix sums, these models solve only 72.45 \pm 8.91% of 512-bit inputs. On maze data, these models were not able to solve any 59x59 mazes in our test set, and we note that the training accuracy is >99% and the accuracy on 33x33 mazes is 93.8%, indicating that training was successful. We are currently completing these experiments on chess data and will include these results in the final version.
> > >
> > > We have added a note about training instability to the main body of the paper, and we will add a more thorough set of plots in the final version.
> > >
> > > The default exit rules results are consistent on chess data as well. We’ve added this plot and confidence vs. iteration plots on each dataset -- see Figures 17 and 18 in the appendix.

---

> > > > ### Comment · Reviewer_Ap3D · 2021-11-23
> > > > **Reply**
> > > >
> > > > Thanks to the authors for this followup, and thanks especially for the inclusion of additional results.
> > > >
> > > > **Comparison to models with run time penalty**: These results are encouraging. I look forward to seeing the results for the chess puzzles as well. Just to clarify, is this for a version of the model *with* recall or *without*? I think the critical comparison here is between their proposed model (recall + progressive loss) and a model where the progressive loss is replaced by a run time penalty (i.e. recall + run time penalty), since the run time penalty is much more analogous to the progressive loss component, in that it is designed to encourage efficient convergence on a solution.
> > > >
> > > > **Specification of progressive loss**: I am curious to hear the authors' perspective on my comments about the progressive loss. Do they agree with my assertions re: computational complexity of their proposed method vs. the simpler alternative method? Have they attempted to test this simpler method and compare to the proposed method?
> > > >
> > > > I appreciate the other additions made to the paper (note about training instability, default exit rule results, time courses of confidence) -- IMO these changes adequately address the respective concerns.

---

> > > > > ### Author Response · Authors · 2021-11-24
> > > > > **Follow up (continued)**
> > > > >
> > > > > The experiments with the run time penalty are models with recall but without the progressive loss. So, yes, as you suggest we are comparing recall+penalty and recall+progressive.
> > > > >
> > > > > We do agree with your run-time analysis. We have now run new experiments with k sampled as previously but setting n = 0. After a thorough investigation with prefix sum data, we see that models train and can generalize to longer input strings but with significantly lower accuracy than with higher values of n. Specifically, DT nets with recall achieve 90.27 \pm 5.95% on 512-bit data with the n=0 loss, and with the higher value of n used in the paper, they achieve 97.12 \pm 1.88%. On the other hand, very preliminary experiments on maze data show less of an advantage to higher values of n.  Thanks for your valuable input.  We conclude that in some cases, n=0 might be a cheap and sufficiently high-performing hyperparameter setting, while higher values provide higher performance in other cases, so n is a hyperparameter worth tuning.  Thus, we are updating our local draft with thorough ablations on values of n and will include these in the final draft.  Note that chess models take significantly longer to train and test, so these results will come in after discussion ends.

---

> > > > > > ### Comment · Reviewer_Ap3D · 2021-11-26
> > > > > > **Reply and score increase**
> > > > > >
> > > > > > Thanks to the authors for this additional followup. I appreciate the additional experiments, which make for a complete picture concerning the paper's primary proposals. I will increase my score accordingly.

---

### Official Review · Reviewer_w7Ak · 2021-11-03

**Correctness:** 4
**Technical Novelty And Significance:** 3
**Empirical Novelty And Significance:** 4
**Recommendation:** 8
**Confidence:** 5

**Main Review:**

The paper is well written and easy to read and understand. It builds directly on recent work by Schwarzschild et al. [2021a, 2021b, 2021c], and is essentially about addressing the limitations of that work on the challenging tasks identified in that work that involve learning from "easy" instances of a task and generalizing to "hard" instances. This is an interesting line of work that could lead to useful insights, and this paper's contributions certainly to make significant progress on the challenges considered.

In particular, the authors have already conducted various analyses to answer follow-up questions that a curious reader might have (and I had) about the working of the proposed techniques. The results of these experiments will be very useful and instructive to readers interested in these problems.

I would like to point out two "weaknesses" of this paper:

1. The first out of the proposed two techniques, the Recall skip connection, appears to be fixing what one may call a mistake made by Schwarzschild et al. in recent work during model design. This is because if one looks at prior literature on models that are stable when unrolled longer than training time, one immediately find this model design, based on theoretical insights from control theory, see Ciccone et al. [A] and the non-autonomous networks therein. This is not to say that the contribution of this paper isn't useful, but it should be put in context of past work that has already established strong theoretical results directly related to the behavior of interest. I find it interesting that in this paper, no conditions on weights were necessary in practice to achieve stability.

2. I'm concerned that the authors are playing fast and loose with the term "thinking" in this line of work. This is very loaded metaphor, and in my opinion every paper that decides to use it must at least include a clear note that "thinking" is just a fancier term for a particular type of processing here, and these networks are not really thinking in any regular sense of the term. Otherwise, we can say that any classifier or detector is also thinking. I realize that this is not just something unique to this paper and some other papers have used such terms without such clarification, but this is only a review of this paper, and I'd appreciate if the authors think about this issue and try to do better than prior work.

[A] Ciccone, M., Gallieri, M., Masci, J., Osendorfer, C., & Gomez, F. (2018). Nais-net: Stable deep networks from non-autonomous differential equations. arXiv preprint arXiv:1804.07209.

**Summary Of The Paper:**

This paper is focused on training networks to solve problems via extrapolation, e.g. solving large mazes by learning to solve small mazes. In order to achieve this, two modifications to recurrent convolutional networks are proposed: a) adding a concatenating skip connection from the input to the recurrent layers that stabilizes extrapolation (Fig. 2), and b) using a modified training method and loss that encourages the network to learn computations independent of current iteration (Alg. 1). Together, these techniques lead to substantial improvements in results on the problems of computing prefix sums, solving 2D mazes and solving chess puzzles.

**Summary Of The Review:**

The paper proposes simple techniques that clearly addresses the limitations of recent work on solving tasks in the easy-to-hard benchmark dataset.

---

> ### Author Response · Authors · 2021-11-19
> **To Reviewer w7Ak**
>
> Thank you for your time and the thoughtful review. We appreciate you bringing [A] to our attention, we have added a reference to [A] and better contextualized our work. Specifically, we note that prior work on classification is very motivating to our project where we take similar architectures into domains that have natural ways to separate easy and hard examples. Also, as opposed to the method in [A], we do not explicitly require that the iterative block be convergent, rather that behavior is learned. With regard to your second point, we agree that “thinking” may be a loaded term, therefore we have clarified in our draft exactly what we mean by thinking.

---

### Author Response · Authors · 2021-11-30
**Overall Author Response**

We appreciate the reviewers' time and constructive feedback. In particular, we appreciate that the reviewers acknowledge how interesting the problem we address is, as well as how compelling and thorough our results are. Multiple reviewers also commented on how well written the paper is -- thank you again!

The main themes of the reviews touched on missing related work, technical clarity and detail, and further experimentation. We have updated our draft to better situate our work among existing works, including adding new references to papers mentioned in the review process. We have also clarified several points in the main body as well as in the appendices. Finally, we performed more tests on our models with different evaluation techniques, and we trained new models with the suggested variations of our proposed loss.

We'd like to highlight that for each reviewer, we answered all the questions asked, and where possible added numerical or graphical results to our updated draft.

---

### Decision · Program_Chairs · 2022-01-20

**Decision:**

Reject

**Comment:**

This is an interesting work, and I urge the authors to keep pushing this direction of research. Unfortunately, I feel like the manuscript, in its current format is not ready for acceptance.

The research direction is definitely under-explored, which makes the evaluation of the work a bit tricky. Still I think that some of the points raised by the reviewers hold, for e.g. the need of additional baselines (to provide a bit of context for what is going on)I understand that the authors view their work as an improvement of the previously proposed DT network, however that is a recent architecture, not sufficiently established not to require additional baseline for comparisons. This combined with the novely of the dataset makes it really hard to judge the work.

The write-up might also require a bit of attention. In particular it seems a lot of important details of the work (or clarifications regarding the method) ended up in the appendix. A lot of the smaller things reviewer pointed out the authors rightfully so acknowledged in the rebuttal and propose to fix, however I feel this might end up requiring a bit of re-organization of the manuscript rather that adding things at the end of the appendix. I also highlight (and agree) with the word "thinking" being overloaded in this scenario.

Ablation studies (some done as part of the rebuttal) might be also a key component to get this work over the finish line. E.g. the discussion around the progressive loss. I acknowledge that the authors did run some of those experiments, though I feel a more in depth look at the results and interpretation of them (e.g. not looking just at final performance, but at the behaviour of the system), and integrating them in the main manuscript could also provide considerable additional insight in the proposed architecture.

My main worry is that in its current format, the paper might not end up having the impact it deserves and any of the changes above will greatly improve the quality and the attention the work will get in the community.